# Theoretical and Experimental Assay of Shock Experienced by Yeast Cells during Laser Bioprinting

**DOI:** 10.3390/ijms23179823

**Published:** 2022-08-29

**Authors:** Erika V. Grosfeld, Vyacheslav S. Zhigarkov, Alexander I. Alexandrov, Nikita V. Minaev, Vladimir I. Yusupov

**Affiliations:** 1Bach Institute of Biochemistry, Federal Research Center of Biotechnology of the RAS, 119071 Moscow, Russia; 2Moscow Institute of Physics and Technology (National Research University), 141700 Dolgoprudny, Russia; 3Institute of Photon Technologies of Federal Scientific Research Centre “Crystallography and Photonics” of Russian Academy of Sciences, Pionerskaya 2, Troitsk, 108840 Moscow, Russia

**Keywords:** laser bioprinting, laser-induced forward transfer (LIFT), yeast, cell death, membrane perturbation

## Abstract

Laser-induced forward transfer (LIFT) is a useful technique for bioprinting using gel-embedded cells. However, little is known about the stresses experienced by cells during LIFT. This paper theoretically and experimentally explores the levels of laser pulse irradiation and pulsed heating experienced by yeast cells during LIFT. It has been found that only 5% of the cells in the gel layer adjacent to the absorbing Ti film should be significantly heated for fractions of microseconds, which was confirmed by the fact that a corresponding population of cells died during LIFT. This was accompanied by the near-complete dimming of intracellular green fluorescent protein, also observed in response to heat shock. It is shown that microorganisms in the gel layer experience laser irradiation with an energy density of ~0.1–6 J/cm^2^. This level of irradiation had no effect on yeast on its own. We conclude that in a wide range of laser fluences, bioprinting kills only a minority of the cell population. Importantly, we detected a previously unobserved change in membrane permeability in viable cells. Our data provide a wider perspective on the effects of LIFT-based bioprinting on living organisms and might provide new uses for the procedure based on its effects on cell permeability.

## 1. Introduction

The rapidly developing and promising laser bioprinting technology [1] is based on the well-known laser-induced forward transfer (LIFT) effect [2]. Using this approach, a wide range of biological materials are already being printed. In general, while bioprinting is more often associated with printing tissues (i.e., using mammalian cells) [3], the bioprinting of microbes is also a developing field [4,5,6,7]. Specifically, the bioprinting of yeast cells has recently been used to create a biocatalyst that produces ethanol [8,9] and for the engineering of bio-hybrid materials [10,11]. Potentially, bioprinting could also be applicable to the growing field of yeast-based biosensors [12]. Recently, it has also been shown that technology based on this approach allows isolating difficult-to-cultivate microorganisms and even microorganisms that could not previously be cultivated [13,14].

For laser bioprinting, a donor plate is usually prepared in advance, consisting of a glass slide with a thin metal coating (Au, Ti, etc.) that absorbs laser radiation, onto which a layer of hydrogel containing biological material is applied.

The laser system is pre-tuned so that the laser radiation is focused on the surface of the absorbing coating. During laser printing, a short laser pulse passes through a transparent plate and is absorbed by the metal film. As a result, the local region of the film and the adjacent thin layer of the hydrogel are rapidly heated to temperatures substantially exceeding the critical temperature of the water. A rapidly expanding high-pressure vapor bubble appears, on top of which a hydrogel jet is formed. Subsequently, a gel microdrop is detached from this jet, which transfers the biological material to the acceptor surface.

To become a truly universal technique, laser bioprinting, in addition to achieving the required frequency of laser shots, accuracy, and stability [15,16,17,18,19], must ensure a minimal negative impact on the transferred biological substance, such as living microorganisms.

During the laser bioprinting process, microorganisms are subjected to various shocks and chronic effects. Shock effects are caused by pulse laser irradiation [20], shock waves [21], temperature change [22], and impulsive pressure gradients associated with acceleration near the donor plate and landing on the acceptor surface [23]. Chronic effects on the microorganisms are due to the effects of the gel and its potential modifications due to laser exposure and nanoparticles from the destroyed area of the absorbing film [24,25].

Knowledge about the effect of laser bioprinting on biological objects is essential to improve the technology. For this, model eukaryotic cells, such as yeast, seem to be a highly useful model to study the impact of the various effects listed above on microorganisms, which are currently lacking.

The goal of this paper was to explore the biological effects of LIFT, including what doses of laser pulse energy cells in the gel are exposed to, what portion of cells are subjected to pulsed temperature heating, and to evaluate the effects of irradiation with transmitted laser pulsed radiation and pulsed heating on yeast cells.

## 2. Results and Discussion

All of the experiments were carried out using a bioprinting system in a LEMS (Laser Engineering Microbial Systems) configuration [26]. This system uses nanosecond-duration pulses of an IR laser to irradiate a glass plate with an absorbing Ti film. Heating of this film causes jets of gel, containing cells, to be shot towards the printing area. This is schematically depicted in Figure 1. In this report, the cells embedded within the gel were yeast belonging to the species Saccharomyces cerevisiae.

### 2.1. Estimation of Localized Heating Effects

To evaluate the potential pulsed heating that microorganisms in the gel layer can undergo during laser transfer, it is first necessary to evaluate the dynamics of the ongoing processes. Figure 2 shows how the speed of the formed jets of gel depends on the energy of the laser pulse. This dependence is linear in the first approximation, while in the operating range of LEMS (18–25 μJ), the initial jet velocity is 40 ± 5 m/s.

The experiments have shown that the action of the laser leads to the formation of holes in the absorbing Ti film of the donor substrate. Figure 3a shows the dependence of the square of the diameter of these holes on the fluence (surface laser energy density). It can be seen that when the graph is presented on a logarithmic scale along the *x*-axis, this dependence appears to be linear.

This result is expected for a laser spot with a Gaussian intensity distribution and short pulse duration. From the Gaussian distribution of the laser fluence in the spot, it is easy to obtain the relation: D^2^ ≈ 4.6·ω_0_^2^·lg(F/F_th_), where D—diameter of the hole, ω_0_—laser beam waist radius, F—laser fluence, F_th_—threshold fluence for the formation of a hole in the film. According to the point of intersection of the linear trend with the axis D^2^ = 0, it can be determined that F_th_ = 0.06 J/cm^2^.

Note that when the gel is applied to the Ti film, the size of the hole in the film decreases significantly: for F = 1 J/cm^2^ from D = 50 µm (Figure 3a) to D = 36 µm (Figure 3b). Obviously, this is due to additional heat losses for heating a thin layer of gel and phase transitions of water. The SEM image of the hole (Figure 3b) distinctly shows that the edges of the Ti film are melted. This circumstance makes it possible to estimate the temperature distribution arising during laser pulse action. Because the laser pulse is very short, the thermal diffusion length ~0.2 μm, where *a* = 6.3·10^−6^ m^2^/s—thermal diffusivity of Ti, is much smaller than the laser spot diameter. Therefore, the temperature profile in the laser spot in the absence of phase transitions will be described by a Gaussian curve of the same shape as the laser radiation intensity distribution (red dashed curve in Figure 3c).

However, at least one phase transition of Ti under pulsed laser action occurred (Figure 3b): the melting boundary of the material is observed at the distance D/2 from the optical axis. Therefore, we can state that along this circular boundary, the Ti film temperature reached the melting temperature T_m_ = 1668 °C. Since a phase transition of the material occurred at this boundary and in the inner part of the hole, the temperature inside this area will also be described by a Gaussian curve (red dotted curve) reduced by the value ΔT = ΔH_m_/C_Ti_, where ΔH_m_ = 358 kJ/kg is the heat of fusion and C_Ti_ = 530 J/(kg·K) is the heat capacity of Ti.

If there were no phase transitions in the inner part of the hole in the Ti film associated with the evaporation of the material, then the maximum temperature in the center of the laser spot, as can be easily estimated, would be T_MAX_ = 6880 °C. Note that this temperature exceeds the boiling point of Ti T_b_ = 3287 °C but does not reach its critical temperature T_c_ = 11,500 °C. Thus, liquid Ti in a spot with a radius of ~10 μm is placed in the so-called metastable region of the phase diagram. Since the stored energy, even in the paraxial region, will not be enough to evaporate all of the Ti material (ΔH_vap_/C_Ti_~16,900 °C, where ΔH_vap_ = 8970 kJ/kg is the heat of vaporization), only a part of it will evaporate. In this case, the temperature in this region will be kept at the level T_b_ = 3287 °C (green curve in Figure 3c). Thus, the proposed simple scheme makes it possible to accurately estimate the temperature distribution profile both outside the hole formed in the film and inside it.

Having obtained the temperature distribution of the absorbing film (Figure 3c), it is now possible to estimate the heat propagation in the gel layer. Figure 4 shows the main stage of gel jet formation. By the end of the laser pulse (τ = 8 ns), a thin (135 nm, where *a_g_* = 0.34·10^−6^ m^2^/s, *a_w_* = 0.14·10^−6^ m^2^/s—thermal diffusivity of glass and water, respectively) layer of glass and gel is heated simultaneously with the Ti film up to the temperature ≥ T_b_/2 ≈ 1650 °C.

By *t* = 300 ns (Figure 4), a gas–vapor bubble is formed near the Ti film. Due to the partial destruction of the Ti film, Ti nanoparticles are formed in the inner part of the spot. They can be located on the surface of the glass slide, inside the bubble, and on the wall of the bubble in a thin layer of gel. At this time, the shock wave (not to scale in Figure 4) is already at a distance of >450 µm from the Ti film. The temperature distribution in the gel in the paraxial region shows that by the end of the laser pulse (*t* = 8 ns), a gel layer ~0.1 μm thick was heated to a temperature of >100 °C. This layer of heated water evaporates into the inner part of the bubble. As a result, the temperature of the gel layer adjacent to the bubble decreases to <100 °C. By 300 ns, the temperature increase ΔT in the gel layer with a thickness of ~1 μm is in the range of 0.1–10 °C. By 10 μs, the temperature jump ΔT in this layer is about 1–2 °C.

Thus, we can conclude that only cells directly adjacent to the Ti film surface should undergo significant heating up to a temperature ≥ 1650 °C. It is important that the time of such temperature exposure does not exceed fractions of a microsecond. Considering that the average size of yeast cells is about 5 µm, the estimated number of cells exposed to ΔT of more than 30 °C in a 200 µm gel layer will be about 5%.

### 2.2. Estimation of Localized Laser Pulse Energy

Another factor that affects the living systems in the gel, which we study in this work, is pulsed laser exposure. Figure 5a shows the dependences of the laser energy transmitted through the Ti film on the energy of the pulse incident on the donor substrate. It can be seen that in the absence of a gel on the Ti film, the amount of transmitted radiation rapidly increases with increasing pulse energy. The application of the gel leads to a significant decrease in the level of transmitted radiation. In addition, although the dependence remains positive, the amount of transmitted energy increases very slowly with increasing pulse energy. In this case, the transmittance coefficient for a pure Ti film increases with increasing pulse energy from 22% to 34% (Figure 5b). In contrast to this, the transmittance in the presence of a gel layer decreases with increasing pulse energy from 15% to 8%.

The fact that part of the laser energy passes through the metal coating of the donor plate is not extraordinary. Firstly, the thickness of the Ti film is only 50 nm and does not absorb 100% of the radiation. Secondly, as mentioned above, the laser pulse results in the partial evaporation and destruction of the Ti film, which leads to an increase in laser energy transmission.

Let us now determine the distribution of the laser radiation field in the gel layer of the donor slide. Figure 6a shows the distribution of laser intensity in the form of isolines in the focusing region in the absence of a donor plate. The shape of the isoline in the waist area may seem close to circular, but it is not the case since the scales on the vertical and horizontal scales differ significantly. In reality, these isolines in the area of laser exposure are ellipsoids strongly elongated in the vertical direction.

Figure 6b shows the distribution of the laser fluence under the action of a laser pulse on a donor plate with a Ti film and a gel layer. The laser energy densities before and after the Ti film differ significantly due to the absorption of part of the laser energy (Figure 6b). As expected, the region with the maximum energy density in the gel layer is located on the optical axis near the Ti film. When moving away from this point in the horizontal direction, the fluence decreases rather quickly from ~0.6 J/cm^2^ in the center of the spot to ~0.1 J/cm^2^ at the edge of the laser spot. With distance from this point in the vertical direction, the fluence decreases very slowly from ~0.6 J/cm^2^ near the Ti film to ~0.4 J/cm^2^ at the lower boundary of the gel layer.

Thus, since the transferred gel microdroplet is formed from a cylindrical volume of gel in the area of laser exposure, it can be estimated that the microorganisms in the gel will be exposed to a laser nanosecond pulse with a fluence of ~0.1–6 J/cm^2^.

### 2.3. Characterization of Yeast Cell Death Using Membrane-Permeability, GFP-Dimming and Viability Assays

#### 2.3.1. Effects of LIFT on Cell Membrane Permeability and Cell Viability

In order to determine whether LIFT causes significant cell death, we tested whether we could detect notable amounts of cells with permeated membranes using propidium iodide staining (PI). PI is a positively charged stain that, usually, cannot enter cells with an intact membrane and penetrates cells with a damaged membrane, staining their nucleic acids. Notably, in some rare cases, it can stain living cells [27,28]. There are also numerous cases of apoptosis-like cell death in yeast where a shock causes cells to cease division but not exhibit immediate membrane permeabilization. A simple experimental indicator of this type of cell death is a non-congruence between the number of PI-positive cells and non-viable cell numbers as determined by colony-forming assays [29,30].

We observed that LIFT does not cause a significant emergence of cells with strong PI staining, as is characteristic of cells killed by boiling in water, which were also submitted to the LIFT procedure (Figure 7a).

However, LIFT did cause the appearance of cells with weak PI staining. This fluorescence was ~200-fold weaker than that characteristic of the heat-killed cells (Figure 7a and Figure 8b) or cells that died due to increased cell death rates during division [31,32]. In order to assay the localization of weak PI fluorescence, we used fluorescent microscopy and confirmed that the localization of the fluorescence was intracellular and cytosolic (Figure 7b).

The fact that the cellular fluorescence of LIFTed cells is much lower than that of the control dead cells suggested that these LIFTed cells might not be dead. This was confirmed via a colony-forming assay, during which we deposited identical numbers of cells (as tested by flow cytometry) onto a solid growth medium that were or were not subjected to LIFT to see what portion of the cells formed colonies (Colony-Forming Units—CFU). This showed that the number of live cells changed only moderately (no more than 15%) (Figure 8c), while more than 80% of the LIFTed cells gained the described weak PI fluorescence (Figure 8a).

Notably, the assay of the number of CFU did indicate that the number of viable cells was somewhat reduced after LIFT (~10% dead cells, vs. less than 4% in the control). If some cells die, it makes sense to assume that more cells experience some form of shock.

#### 2.3.2. Comparing Effects of LIFT, Laser Irradiation and Heating

As discussed above, the two most likely types of shock that could cause cellular stress are laser irradiation and heat. We tested whether laser irradiation intensities (without LIFT) that were lower, equal to, or higher than those calculated above had any effect on cells and observed no change in the cell membrane permeability (Figure 8d) or the numbers of viable cells (data not shown).

To determine whether the cells experienced heat shock during LIFT, we tested whether the LIFTed cells changed the level of production of the Hsp70-family chaperone Ssa1. This chaperone increases its level in response to heat shock [33], as well as other types of cellular stress [34]. We predicted that if the LIFT procedure caused rapid heat shock, then some cells might increase the level of a green fluorescent protein-tagged version of this protein, Ssa1-GFP.

Cells harboring Ssa1-GFP were subjected to LIFT. As before, there was no noticeable emergence of strongly PI-fluorescent cells, and most cells gained weak PI-fluorescence (data not shown). Interestingly, we observed no cells with increased GFP signal (Figure 9a). However, a noticeable fraction of cells (5–12% vs. ~3.5% in the control) lost GFP-fluorescence (Figure 9b). If the cells were PI-positive, this could be explained by leakage of the protein out of the cell; however, most of the cells displayed weak PI-fluorescence. The most likely explanation for the loss of GFP fluorescence is some combination of temperature-induced denaturation and the degradation of the Ssa1 portion of Ssa1-GFP because GFP itself is reported to be relatively stable to in vitro heat-induced denaturation. Specifically, according to the literature [35,36], while GFP loses ~50% of its fluorescent intensity after 5 min of heating at 70 °C, it retains the remaining 50% even after 30 min of the same treatment.

In order to ascertain how Ssa1-GFP behaved under genuine heat shock conditions, we assayed the response of cells to a 1 min heat shock (Figure 9c,d). Importantly, for the cells prepared in the standard manner (high-density culture), we observed no increase in GFP-fluorescence, probably due to the fact that a high-density culture in glucose already has an elevated level of Ssa1 compared to a low-density rapidly dividing culture. However, we did observe that those cultures exposed to 70 °C for 1 min almost completely lost GFP fluorescence. Notably, ~50% of these cells still exhibited a relatively low PI signal, suggesting that their membrane barrier function was mostly intact. Both of these behaviors are reminiscent of the effects we observed for LIFTed yeast cells (Figure 9c,d).

On the other hand, the cells that were grown to lower densities did exhibit an increase in Ssa1-GFP levels in response to a 1 min 50 °C heat shock but did not do so in response to LIFT (Appendix A).

Thus, we can conclude, based on the reduction in the number of viable cells, as well as the disappearance of GFP fluorescence, that a small but statistically significant portion of yeast cells die during LIFT. Since we also demonstrate that genuine heat shock can result in cell death accompanied by GFP fluorescence loss without the emergence of strong PI fluorescence, a likely explanation is that heat shock might be the reason for the observed cell death during LIFT. However, it seems that this shock is so brief that if it is not immediately lethal, it does not cause a traditional heat shock response.

## 3. Materials and Methods

### 3.1. Laser System for Bioprinting

The studies were carried out on an experimental apparatus for the Laser Engineering of Microbiological Systems (LEMS) [26], the schematic diagram of which is shown in Figure 1. A laser pulse from a YLPM-1-4x200-20-20 pulsed fiber laser (NTO “IRE-Polus”, Fryazino, Russia) passed through a beam shaper (2 in Figure 1) and with the help of the LscanH-10-1064 Galvano scanning head (AtekoTM, Moscow, Russia) with an F-theta lens SL-1064-110-160 (Ronar-Smith, Singapore) with a focal length of 160 mm was focused on the surface of the absorbing coating of the donor plate (5). The laser pulse duration was τ = 8 ns, the wavelength λ = 1064 nm, and the laser pulse energy E = 14–35 μJ. The intensity of the laser beam had an almost Gaussian profile with M^2^ < 1.3, and the radius of the laser spot (at the intensity level of 1/e^2^) in the waist was ω_0_ = 15 ± 1 μm. The fluence in the laser spot in this case F[J/cm^2^] = E/(πω_0_^2^) ≈ 0.14·E[μJ] = 2.0–5.0 J/cm^2^, and the maximal fluence at the optical axis is equal to 2·F.

### 3.2. Donor Substrate and Gel Application

Glass slides (Menzel Glaser, Braunschweig, Germany) (26 mm × 76 mm × 1 mm) were used as the donor substrate, on the surface of which Ti was deposited by magnetron sputtering using a VSE-PDV-DESK-PRO installation (OOO Vacuum Systems and Electronics, Novosibirsk, Russia). The film surface was studied by atomic force microscopy on a Solver Pro M complex (ZAO Nanotechnology MDT, Moscow, Russia) in order to determine the thickness of the Ti film, which was 50 ± 10 nm.

Immediately prior to laser transfer, a layer of empty gel or gel mixed with yeast was deposited on the surface of the Ti film of the donor substrate (200-μm thick). The gel was an aqueous solution of 2% hyaluronic acid (*M*_w_ = 70 kDa, Contipro Pharma, Dolní Dobrouč, Czech Republic).

Opposite the donor substrate with the gel, a 12-well plate (Costar Corning, Corning, NY, USA) was placed and fixed on an automated moving platform. For visual control of the process of formation and the transfer of the gel jets and drops, a Fastcam SA-3 high-speed camera (Photron, Tokyo, Japan) with frontal illumination from a continuous laser with a wavelength of 630 nm was used. After laser exposure, the Ti films were examined using a PHENOM ProX scanning electron microscope (Phenom-World, Eindhoven, The Netherlands) and an HRM-300 Series optical 3D microscope (Huvitz, Anyang-si, Gyeonggi-do, Korea).

### 3.3. Pulse Energy Measurement

The laser energy was calibrated using a Gentec QE8SP-MT-INT pyroelectric detector (Gentec-EO, Quebec City, QC, Canada). This detector was also used to estimate the transmitted energy and transmission coefficients through the Ti film of the donor substrate with and without a gel layer. For this, the detector head was installed coaxially with the optical axis under the donor plate at a distance of 7 cm. The choice of the distance was determined by two factors: (1) the intensity of the laser pulse on the detector surface was below the admissible threshold, and (2) all the radiation transmitted through the donor substrate fell onto the working detector area. The surface of the detector was covered with a thin glass plate to prevent the gel microdroplets from falling on it.

### 3.4. Estimation of Temperature Fields

As the reference points of the temperature distribution in the plane of the Ti-absorbing film under pulsed laser irradiation, the boundaries of the melting front of the material were used, as determined from the SEM images of the donor substrate sections. To estimate the temperature distribution during the heating of the thin layers of glass and gel adjacent to the absorbing Ti film, we used the numerical solution of the nonstationary heat conduction equation [37] using the tabulated values of the thermophysical parameters for Ti, glass, and water.

### 3.5. Laser-Induced Forward Transfer (LIFT) Procedure

*Saccharomyces cerevisiae* of the BY4741 strain (MATa his3Δ0 leu2Δ0 met15Δ0 ura3Δ0), as well as the Ssa1-GFP derivative of this strain [38], were used for these experiments. Ssa1 is an Hsp70-family heat shock protein that is known to bind to protein aggregates and increases in abundance after heat shock and other perturbations. Yeast were inoculated in 10 mL of YPD medium (1% yeast extract, 2% peptone, and 2% glucose, all *w*/*v*) in sterile 50 mL Falcon tubes and incubated overnight at 30 °C and 250 rpm. The cells were collected by centrifugation (1000× *g*, 2 min) at room temperature and resuspended in 2% HA solution (1:3 ratio) to a final concentration of 1.5% HA. The gel was then spread onto the surface of the Ti-covered donor plate and placed into the experimental apparatus, as illustrated in Figure 10. The untreated yeast cells, as well as cells mixed with HA gel (controls), were placed in 2 of the wells before the start of the experiment. The remaining 10 wells (with 250 μL H_2_O) were used for depositing cells by LIFT at different laser energies (14–35 μJ). About 60–90 impulses were usually required to deposit the required amount of cells (10,000) for consequent flow cytometry, based on the number of cells in the original cell suspension.

### 3.6. Determination of Membrane Permeability

For the detection of cells with permeabilized membranes (which are usually dead), cell solutions obtained after LIFT were stained with propidium iodide (PI) (2 μg/mL) for 30 min in sterile distilled water. Flow cytometry was performed in 96-well plates (N-701011 96, Wuxi NEST Biotechnology Co., Wuxi, Jiangsu, China) using a Cytoflex S cytometer (Beckman Coulter, Brea, CA, USA) equipped with a 488 nm laser. Where applicable, cell fluorescence was measured in the green channel (525/40 nm) (for GFP fluorescence detection) and red channel (690/50 nm) (for PI fluorescence detection). Wide-field fluorescent microscopy of dead cells on a solid agar pad [39] was performed using a Nikon ECLIPSE Ti inverted microscope (Nikon, Tokyo, Japan) with an LED excitation light source (532 nm) and a 624/40 filter for PI fluorescence detection.

### 3.7. Determination of Cell Viability

To determine the number of viable cells in a suspension, we determined the number of colony-forming units (CFU), i.e., the number of units in a cell suspension that can form a colony, when this suspension is deposited onto a solid growth medium. Suspensions of the LIFTed cells were assayed by flow cytometry to determine cell numbers using separate aliquots. After that, the cell suspensions were spread onto solid YPD medium (2% agar *w*/*v*) in an amount calculated to produce an expected 300 CFU if 100% of the cells were viable. For the graph in Figure 8c, the % of dead cells was calculated as:$$100\%-\left(\frac{\mathrm{average}\;\#\;\mathrm{of}\;\mathrm{colonies}\;\mathrm{in}\;\mathrm{experiment}}{300}*100\%\right)$$

### 3.8. Heat Shock Experiments

The cell suspensions (in distilled water) were placed for 60 s into a Termit metal heating block (DNK-Tekhnologii, Moscow, Russia) set to a specific temperature. After this, the cells were allowed to remain at room temperature for 30 min (with or without PI), after which they were plated onto solid YPD to determine the number of CFUs, or were analyzed on the flow cytometer to determine cell fluorescence. The presence of PI during the incubation did not affect the number of CFU.

### 3.9. Cell Irradiation with Direct Laser Pulse

Since some of the radiation in laser bioprinting passes through the absorbing film of the donor substrate, it can have a direct effect on the cells in the gel layer. Therefore, special experiments were carried out to study the effect of direct irradiation with laser pulses on cells. The doses of laser exposure F_1_–F_3_ were chosen based on the results of measurements of the energy transmitted through the donor substrate with a gel layer (Section 3.3), corresponding to the dose of irradiation of cells in the gel F_g_. Three doses were chosen: (1) lower than the dose F_g_ at bioprinting, (2) at the same level as F_g_, and (3) higher than F_g_.

A cell suspension was applied in a thin layer to the bottom of the well of a 96-well plate (layer thickness ~1 mm). Then, the entire suspension was irradiated with successive laser pulses. The characteristics of the pulses (average energy density, pulse energy, and waist of the laser spot) are shown below:F_1_ = 93 mJ/cm^2^ (E_1_ = 33 µJ, ω = 106 µm)(1)
F_2_ = 283 mJ/cm^2^ (E_2_ = 100 µJ, ω = 106 µm)(2)
F_3_ = 910 mJ/cm^2^ (E_3_ = 100 µJ, ω = 60 µm)(3)

## 4. Conclusions

Based on the experimental studies and modeling of thermal processes, the temperature distribution in the laser spot on the donor plate was determined. This analysis suggests that only 5% of cells in the gel layer that are adjacent to the absorbing Ti film might undergo significant heating. In this case, the duration of such temperature exposure does not exceed fractions of microseconds. It has been established that during pulsed laser action, microorganisms in the gel layer are irradiated with an energy density of ~0.1–6 J/cm^2^.

In terms of the biological effects of LIFT, we have obtained data that demonstrate that LIFT kills only a moderate share of the yeast cells at the tested laser intensities (no more than 20%, ~10% average) thus most cells remain viable. However, these cells exhibit some increase in the permeability of their cellular membranes, as evidenced by increased staining with PI, albeit with an intensity two orders of magnitude lower than for heat-killed cells. This phenomenon requires further study in terms of its severity, duration, etc. It might also have practical value if it can be shown that the LIFTed cells are more permeable to small or large molecules (such as drugs or nucleic acids). Possibly the phenomenon is related to the recently described laser optoporation. It is based on the laser plasmon heating of metal particles located near a cell membrane [40]. This leads to nondamaging poration of the membranes of mammalian cells and thereby facilitates the diffusion of various macromolecules, such as DNA, which can be used, for cell transfection [41].

Importantly, our observations do not hamper the practical use of LIFT because the majority of the cells subjected to LIFT remain viable.

While we were unable to definitively show the induction of a specific heat shock response in LIFTed cells (monitored via assaying the levels of the GFP-tagged chaperone Ssa1), we showed that the LIFTed cells, as well as the cells heat shocked in an incubator, can exhibit loss of GFP fluorescence. Since, during LIFT, the number of cells that lose GFP fluorescence and the number of non-viable cells assessed by CFU assays are similar, we think it likely that LIFT kills a moderate share (5–20%) of cells via short severe heat shock.

These results are the first to demonstrate some effect of the laser bioprinting procedures on the membrane properties of viable cells and also demonstrate the effects of the LIFT procedure on cell death and protein behavior inside dying cells. This provides a more quantitative understating of the effects of laser bioprinting procedures on living organisms and might provide novel uses for bioprinting in the future.

## Figures and Tables

**Figure 1 ijms-23-09823-f001:**
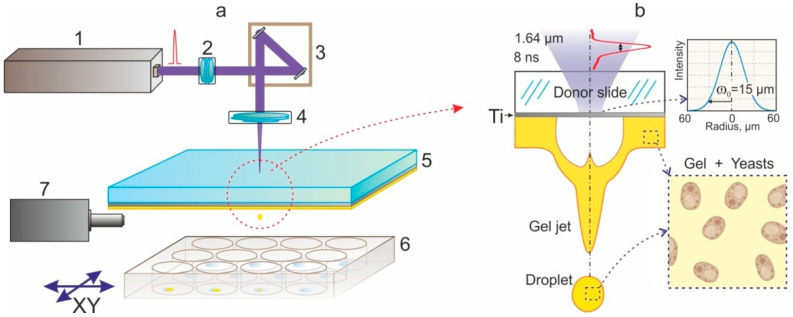
Schematic of the bioprinting setup. (**a**)—Sketch of experimental setup for laser bioprinting. 1—pulsed laser (1064 nm, 8 ns), 2—beam shaper, 3—galvanoscanner, 4—objective, 5—donor substrate with gel layer, 6—bioplate, 7—high-speed camera. (**b**)—schematic representation of a part of a donor plate with a formed jet and a separated microdroplet of gel with yeast cells. The shape of the laser pulse and the distribution of the laser intensity on the surface of the absorbing Ti film are shown.

**Figure 2 ijms-23-09823-f002:**
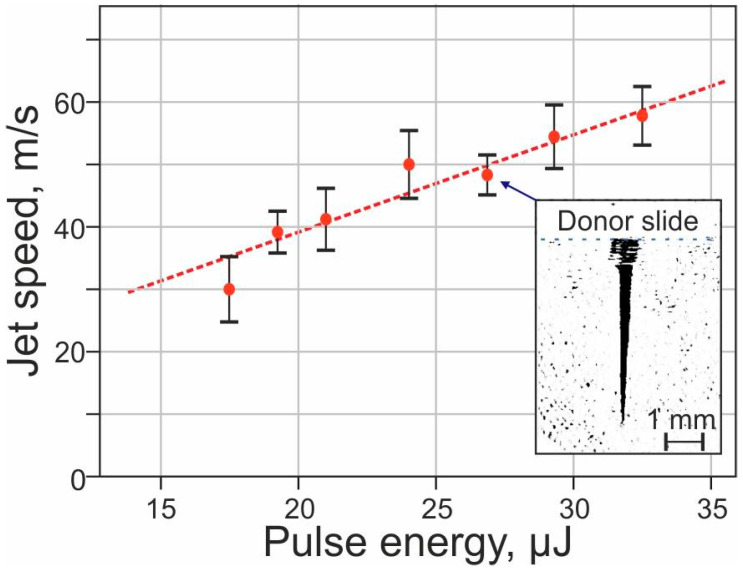
Dependence of gel jet velocity on laser pulse energy. The inset shows an example frame from a high-speed video of a gel jet formed 100 μs after laser pulse absorption.

**Figure 3 ijms-23-09823-f003:**
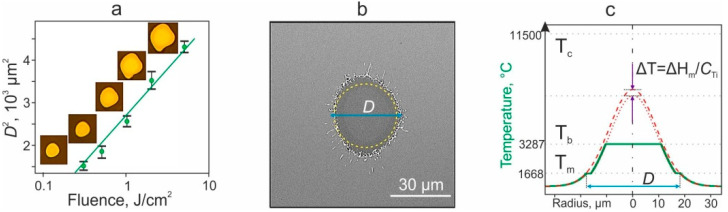
Effect of laser pulses on an absorbing Ti film of a donor substrate. (**a**)—Dependence of the square of the hole diameter D in the Ti film of the donor slide on the laser fluence in the absence of a gel layer on the surface of the Ti film. The inserts show the corresponding optical images of the holes. Linear trend shown. (**b**)—SEM image of a hole in the Ti film with melted edges when the gel layer was previously deposited on the surface of the Ti film. The yellow dotted line shows the area of the laser spot. The effective diameter of the formed hole D is shown. (F = 1 J/cm^2^). (**c**)—Green line shows the distribution of the maximum temperature in the Ti film under laser exposure in the presence of melting and vaporization of Ti. The dashed red Gaussian curve corresponds to the temperature distribution in the absence of phase transitions. The dotted red curve in the inner part of the hole corresponds to the temperature distribution in this area if vaporization of Ti does not occur. This curve is shifted down relative to the dashed curve by ΔT due to energy expended during melting. Melting (T_m_), boiling (T_b_), and critical (T_c_) temperatures of Ti are shown.

**Figure 4 ijms-23-09823-f004:**
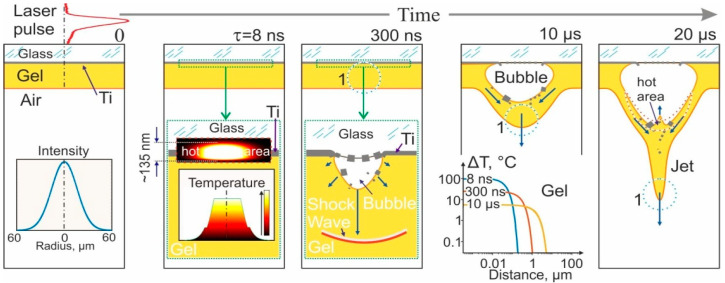
Main stages of gel jet formation during laser pulsed heating of the Ti film of the donor slide after an 8 ns laser pulse. By the end of the pulse with τ = 8 ns, a thin (total ~135 nm) layer of glass and gel is heated simultaneously with the Ti film up to the temperature ≥ T_b_/2 ≈ 1650 °C. By *t* = 300 ns, a gas-vapor bubble is formed near the Ti film and begins to expand. Ti nanoparticles are visible in the region of the laser spot, inside the bubble, and in the thin gel layer on the surface of the bubble. A shock wave leaves the area of laser impact (not to scale). By *t* = 10 µs, a gel jet begins to form. The inset shows the temperature jump ΔT distribution in the gel in the paraxial region at different times. The distance from the bottom wall of the bubble is indicated. By *t* = 20 µs, the jet elongates, and a counter jet is formed. The heated liquid layer is mainly concentrated on the surface adjacent to the bubble (the area is highlighted by the red dotted curve). The blue dotted curve 1 marks the region of the jet from which a microdrop is subsequently formed, which transfers to the acceptor surface.

**Figure 5 ijms-23-09823-f005:**
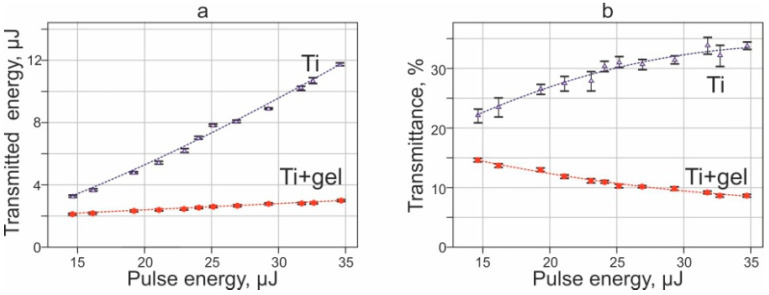
Dependences of the transmitted laser energy (**a**) and transmittance of donor slide (**b**) in the cases of a pure Ti film without gel layer (Ti) and a gel layer on a Ti film (Ti + gel).

**Figure 6 ijms-23-09823-f006:**
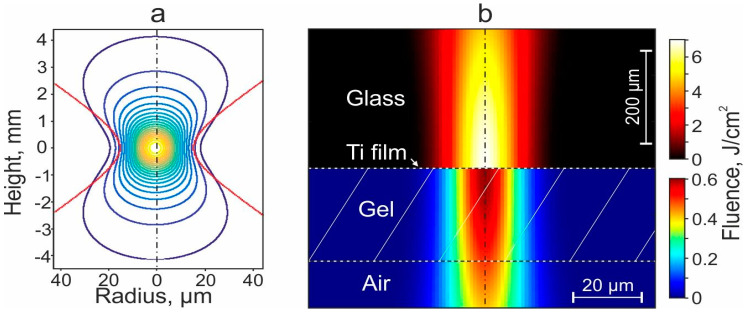
Distributions of laser intensity and fluence. (**a**)—Distribution of the laser intensity in the focusing region in the absence of a donor plate. (**b**)—Distribution of laser fluence during laser pulse action on a donor plate with Ti film and gel layer. For distributions in glass (above the Ti film) and in the gel layer their color palettes are shown.

**Figure 7 ijms-23-09823-f007:**
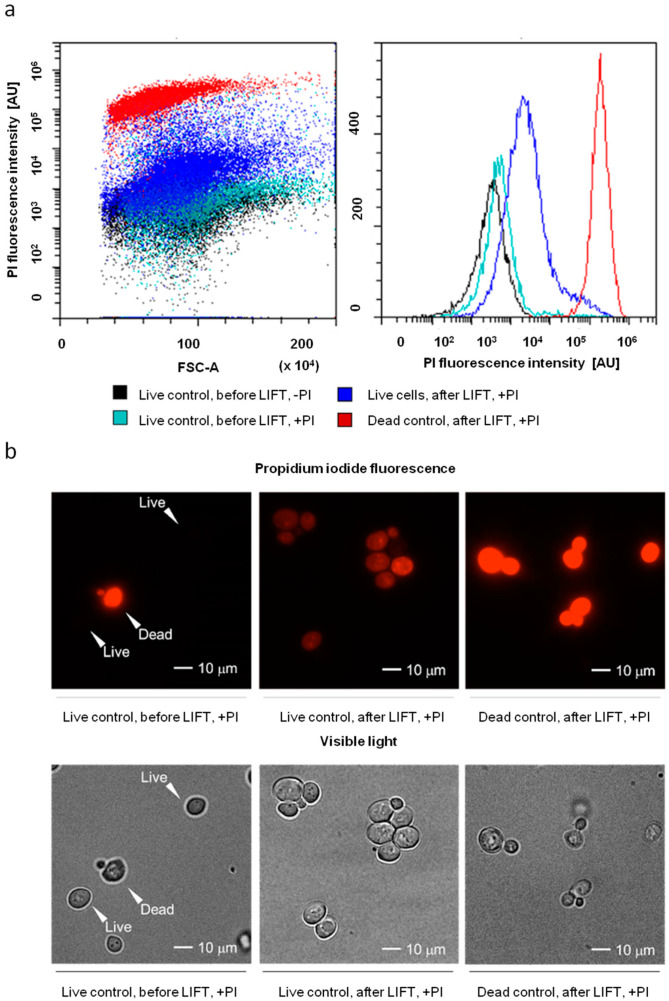
Emergence of weak cytosolic PI-staining after LIFT. (**a**) Flow cytometric analysis of PI-fluorescence (**b**) Fluorescent and DIC microscopy of cells.

**Figure 8 ijms-23-09823-f008:**
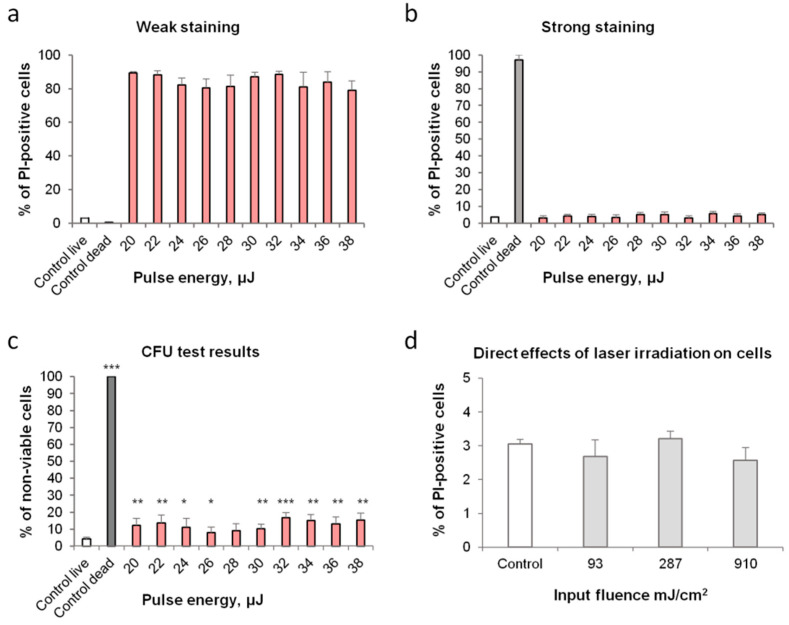
LIFT causes weak PI staining in most cells while resulting in minor, yet significant amounts of cell death not associated with strong PI staining. Laser irradiation alone does not have these effects. (**a**)—Share of cells with weak PI staining in a population of LIFTed cells vs. the laser pulse intensity (**b**)—Share of cells with strong PI staining in a population of LIFTed cells vs. the laser pulse intensity (**c**)—Dependence of % of dead cells in a population of LIFTed cells (see Materials and methods) vs. the laser intensity (*p* ≤ 0.1 (*), *p* ≤ 0.05 (**), *p* ≤ 0.01 (***)), Student’s *t*-test) (**d**) Dependence of % of dead cells in direct laser irradiation vs. the laser intensity.

**Figure 9 ijms-23-09823-f009:**
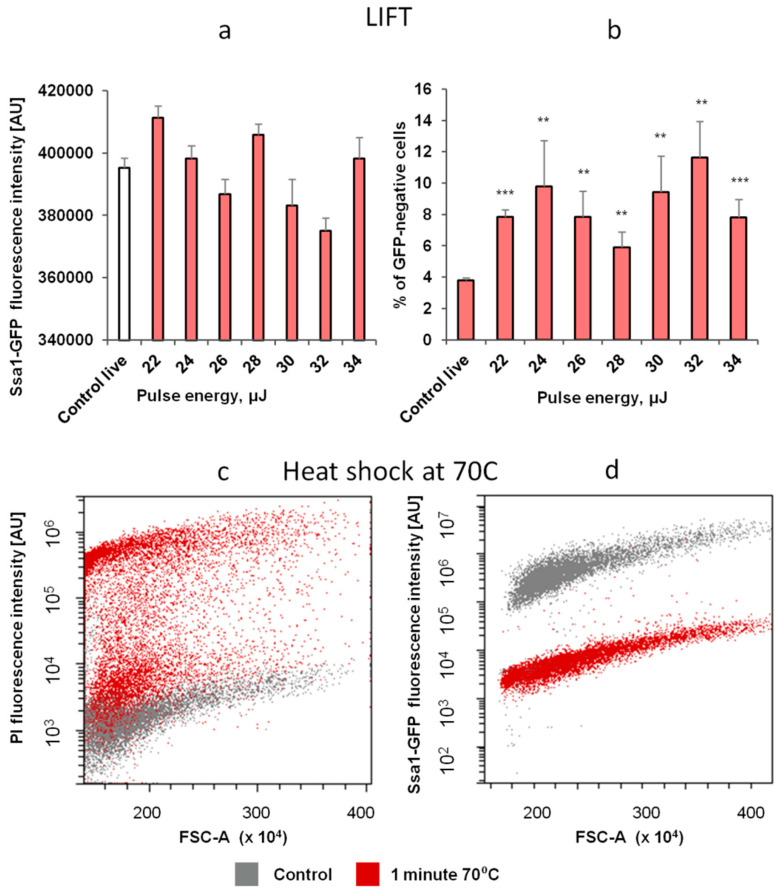
LIFT of Ssa1-GFP cells shows no rise of GFP signal but GFP dimming in a small population of cells, which is also observed in a larger fraction of cells during short heat shock at 70 °C. (**a**) Relation between level of median GFP fluorescence and the laser pulse intensity (in PI-negative cells with noticeable GFP fluorescence) (**b**) Relation between share of cells with diminished GFP signal and the laser pulse intensity (*p* ≤ 0.05 (**), *p* ≤ 0.01 (***), Student’s *t*-test) (**c**,**d**) Cytometric analysis of PI (**c**) and GFP (**d**) fluorescence of yeast cells producing Ssa1-GFP after 1 min treatment with the indicated temperature.

**Figure 10 ijms-23-09823-f010:**
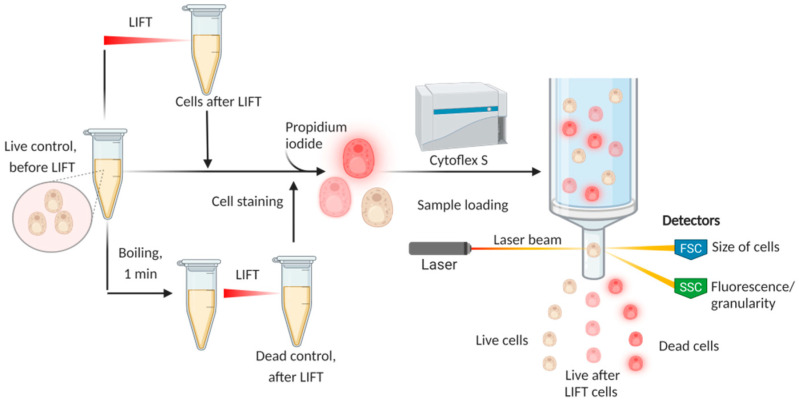
Schematic of the experimental setup. Created with BioRender.com.

## Data Availability

Not applicable.

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
