# Peer review of "Theoretical and Experimental Assay of Shock Experienced by Yeast Cells during Laser Bioprinting"

_ijms, 2022, doi:10.3390/ijms23179823_

Round 1

Reviewer 1 Report

In this article, authors observed heat shock effect of yeast cells during laser-induced forward transfer bioprinting process. And they suggested the suitable energy density to use laser-induced forward transfer. However, in this study, they only tested yeast, relatively strong cells and they did not confirmed the long term functionality of the cells. In order to evaluate the performance of their bioprinting method, it is necessary to check cell viability and functional maintenance by utilizing sensitive cells such as stem cells. In addition, it is thought that it would be more appropriate to submit to the ‘Bioengineering’ of MDPI than to this journal (International Journal of Molecular Science).

Author Response

Point 1: In this article, authors observed heat shock effect of yeast cells during laser-induced forward transfer bioprinting process. And they suggested the suitable energy density to use laser-induced forward transfer. However, in this study, they only tested yeast, relatively strong cells and they did not confirmed the long term functionality of the cells. In order to evaluate the performance of their bioprinting method, it is necessary to check cell viability and functional maintenance by utilizing sensitive cells such as stem cells. In addition, it is thought that it would be more appropriate to submit to the ‘Bioengineering’ of MDPI than to this journal (International Journal of Molecular Science).

Response 1: We chose to conduct our tests with yeasts because the biologist members of our research team are interested in bioprinting with microbes. Specifically, they specialize in yeast, which are commonly used as a chassis organism for the production of various bio-sensors. Notably, the title of our paper notes yeast specifically, we feel that study of other types of cells deserves separate treatment by specialists (for instance in stem cells) in a different work. Notably, LIFT has been used to print with other types of cells and these works are referenced in the introduction. 

We thank the reviewers for bringing up the issue, because we were able to find several papers that we previously missed and have now included additional references concerning bioprinting with yeast.

On a technical note, currently, the instrumentation for LIFT is experimental and is located at a distance from the biological laboratory. Because of this, yeast seemed like a safer choice for conducting work in these conditions. 

The long term functionality of cells is an undefined quantity with yeast. unlike non-dividing somatic cells, or slowly dividing cells. However, because the LIFTed cells effectively survive and produce progeny, they seem, in a replicative sense, to be “functional”. Both theoretically and practically, this parameter is the most important one for bioprinting with (yeast) microbe cells, although, we acknowledge that this is dependent on the specific goal in mind. 

The paper was submitted to IJMS due to a call for the special issue in bioprinting, thus we feel that the material is in complete accordance with this topic.

Reviewer 2 Report

Generally an interesting and well-conducted study on the effect of the laser on yeast cells during LIFT bioprinting. The intro gives a short but to the point background for the topic, starting with nicely explaining the LIFT technique; it is missing some in-depth information about the reason for choosing yeast cells though. The results are presented in an ordered manner and explained nicely, the corresponding discussion sometimes lacks a bit of depth though and especially fitting citations.

Abtract

Wording: Can it really be said, that “little about the stresses experienced by cells during LIFT”? In my opinion (and also going by your introduction, esp. line 49-55), quite a lot is known already…

Results

Line 113: The SEM should refer to Fig 3b, I guess?

Line 133-139: You’re talking about fig 4 b & c, but fig 4 is not subdivided?

Also, did you confuse figure 4 and 5 in the text (from line 144)? Please just double check all your figure references…

Line 167: “should”? From the text I would gather that you mean “it is expected that only those cells do undergo heating”. Or do you really mean “this is the result we want to achieve”?

Line 221-223: Please reconsider this statement. A lot of cells, when they age (or differentiate, though this is not applicable to yeasts) lose their ability to proliferate, they are nevertheless ALIVE though, so there’s absolutely no reason why PI should stain them. Furthermore at least source 20 is specifically talking about apoptosis being a process in which membrane permeabilization is not the first step, so if you specifically want to emphasize, that some “about to be dead” cells are still not stained for PI in your results maybe go by that route of explanation?

Line 230/231: Fig 7 b shows cells in the same size, just some of them are brighter than the others - which is exactly the point you want to make, yes, I get that, however, I find it very difficult to reason “intracellular and cytosolic” from a signal that in size and shape very closely resembles the dead cells.

M&M

Line 371: Distilled water? Really? Distilled water generally kills cells via differences in osmotic pressure – or is there something I don’t know about yeasts?

Literature

Line 34/35, source 4: This source appears to be an advertisement for bioprinting in space special issues, not a proper research article, not even a review. Please cite another source for the statement

Line 47/48, source 8: While it is a great review on the technology, as far as I can see, this manuscript does not talk about microorganisms anywhere. There’s a lot about cells, but not about microorganisms.

Source 19: is not accessible via a normal google search

Author Response

Abstract

Q: 1. Wording: Can it really be said, that “little about the stresses experienced by cells during LIFT”? In my opinion (and also going by your introduction, esp. line 49-55), quite a lot is known already…

A: 1. We agree with the reviewer that the statement is misleading, we have changed the wording to: However, little is known about which stresses imposed by LIFT can cause cell death and other deleterious effects.

Results

Q: 2. Line 113: The SEM should refer to Fig 3b, I guess?

A: 2. We have rechecked the reference to the figure and substituted it for the appropriate one.

Q: 3. Line 133-139: You’re talking about fig 4 b & c, but fig 4 is not subdivided?

Also, did you confuse figure 4 and 5 in the text (from line 144)? Please just double check all your figure references…

A: 3. We have rechecked the reference to the figure and substituted it for the appropriate one.

Q: 4. Line 167: “should”? From the text I would gather that you mean “it is expected that only those cells do undergo heating”. Or do you really mean “this is the result we want to achieve”?

A: 4.  We thank the reviewer for this comment and we have included the suggested phrasing: Thus, we can conclude that only cells directly adjacent to the Ti film surface are expected to undergo significant heating up to a temperature…

Q: 5. Line 221-223: Please reconsider this statement. A lot of cells, when they age (or differentiate, though this is not applicable to yeasts) lose their ability to proliferate, they are nevertheless ALIVE though, so there’s absolutely no reason why PI should stain them. Furthermore at least source 20 is specifically talking about apoptosis being a process in which membrane permeabilization is not the first step, so if you specifically want to emphasize, that some “about to be dead” cells are still not stained for PI in your results maybe go by that route of explanation?

A: 5. The issue of what is alive and what is not, although seemingly simple, is a difficult one, requiring a long discussion. While many human cells do not replicate and are alive, yeast, when presented with the opportunity (i.e. rich growth medium) always divide, unless something in them has been damaged. I.e. yeast do not (at least to our knowledge) have fully functional non-dividing forms in conditions where division normally proceeds. Interestingly, this has been described for mycobacteria, for instance (dormant forms). 

Most importantly, in yeast, such forms have not been shown to emerge due to shocks, or at least, no work we know of has demonstrated that a large portion of cells that seem replicatively dead can be “reawoken”. Aged yeast cells can stop division and spend a prolonged time in a senescent/quiescent state before death, however the general consensus is that they are physiologically still alive, but, replicatively, dead and this is considered to be a form of cell death.

We have changed the wording of the passage, we hope the reviewer finds the new version to be more clear.

Q: 6. Line 230/231: Fig 7 b shows cells in the same size, just some of them are brighter than the others - which is exactly the point you want to make, yes, I get that, however, I find it very difficult to reason “intracellular and cytosolic” from a signal that in size and shape very closely resembles the dead cells.

A: 6. The Intracellular and cytosolic descriptors aim to emphasize that the cells do not glow on the outside (cell wall or membrane). Dead cells (although brighter) also glow with a similar pattern. Our chief reason for doing the microscopy experiment was to determine whether dead cells and live-weakly staining cells have different patterns of fluorescence. As the reviewer understood correctly, they do not. I.e. the difference is only in the signal brightness. This allows us to conclude that LIFT induces some intermediate form of membrane permeability, not, for instance, dye binding to the cell wall or membrane. In general a distribution that is cytosolic means that the cell glows, and no specific type of organelle staining is observed (nuclear, vacuole, puncta, ER, mitochondria etc.)

M&M

Q: 7. Line 371: Distilled water? Really? Distilled water generally kills cells via differences in osmotic pressure – or is there something I don’t know about yeasts?

A: 7. We thank the reviewer for his careful reading of the manuscript. Distilled water does not kill yeast cells, and is commonly used for resuspending and storing yeast during microbiological manipulations. This is because yeast have a cell wall, which prevents cells from dying (via bursting) or even experiencing strong stress during hypo-osmosis. The yeast cell simply pushes against the cell wall with a little bit more force.  Hyposmosis is relatively common in the life of a microbe that lives on the surfaces of plants (rain water is rather hypoosmotic for instance) and its commonality is probably one of the important reasons for having a cell wall in the first place. 

So, in short, resuspension in distilled water is completely normal in yeast microbiology. 

The reviewer might enjoy a fun side fact that, amazingly, distilled water with added glucose does kill yeast quite efficiently, which is known as sugar-induced cell death, but this is most likely a maladaptive signalling response, not a purely physical phenomenon.

Literature

Q: 8. Line 34/35, source 4: This source appears to be an advertisement for bioprinting in space special issues, not a proper research article, not even a review. Please cite another source for the statement

A: 8. We thank the reviewer for his attention to the sources, we have rechecked the reference and substituted it for the appropriate one. 

Q: 9. Line 47/48, source 8: While it is a great review on the technology, as far as I can see, this manuscript does not talk about microorganisms anywhere. There’s a lot about cells, but not about microorganisms.

A: 9. The reviewer is correct, this is a case of incorrect placement, we have moved the noted references in front of the mentioning of the microorganisms.

Q: 10. Source 19: is not accessible via a normal google search

A: 10. We checked and the paper is available on pubmed and via its doi. Possibly the reviewer meant a different reference?

Reviewer 3 Report

This study theoretically and experimentally explores the levels of laser pulse irradiation and pulsed heating experienced by yeast cells during LIFT. It has been found that only 5% of the cells in the gel layer adjacent to the absorbing Ti film should be significantly heated for fractions of microseconds. In a wide range of laser fluences, bioprinting kills only a minority of the cell population. Importantly, the authors detected a previously unobserved change in membrane permeability in viable cells. Some questions need to be answered by the authors.

(1)  With regard to bioprinting, mammalian cells are generally used. Why the yeast cells, but not the mammalian cells were chosen by the authors? Compared to the mammalian cells, the yeast has a protective cell wall on its surface.

(2)  In the part of the conclusion, four paragraphs were listed. Some details are not necessary in this section. It is suggested that the current conclusion should be more condensed. In addition, the writing of the introduction should also be improved.

(3)  In order to test the effect of laser on the cell viability, it is suggested that a specific structure model containing yeast cells should be printed and showed in the article.

Author Response

Q: 1. With regard to bioprinting, mammalian cells are generally used. Why the yeast cells, but not the mammalian cells were chosen by the authors? Compared to the mammalian cells, the yeast has a protective cell wall on its surface.

A: 1. We thank the reviewers for bringing up the issue, because we were able to find several papers that we previously missed and have now included additional references concerning bioprinting with yeast. Microorganisms are also used for bioprinting (see added refs 4-14), for instance in various applications involving microbes producing biosensor-elements. Because microbes are easier to cultivate and gene-engineer, and yeast is one of the best microbes in this area, it seems fitting to use yeast as a model organism, if the goal of the bioprinting will be to work with yeast-based sensors. Importantly, we feel our data might stimulate similar work on mammalian cells. The yeast cell wall might not affect resistance to heating, which seems to be highly relevant in our work. The novel cell permeabilization phenomenon we plan to study more thoroughly in future work, possibly both in yeast and mammalian cells. Also, because this work was a collaboration between laser physicists and biologists, the researchers working on yeast were willing to collaborate on short notice and quite rapidly. Also, yeast was quite convenient to transport quickly between the laser lab and the cell culture lab.

Q: 2. In the part of the conclusion, four paragraphs were listed. Some details are not necessary in this section. It is suggested that the current conclusion should be more condensed. In addition, the writing of the introduction should also be improved.

A: 2.  In our opinion, the conclusion section is not overly long, and for the ease of the reader, we prefer to repeat the main quantitative points. Thus we prefer to leave it as is.  For the introduction, we have added details on the bioprinting of yeast and microbes in general and edited the text. We hope it is now improved. We thank the reviewers for helping us improve the manuscript.

Q:3. In order to test the effect of laser on the cell viability, it is suggested that a specific structure model containing yeast cells should be printed and showed in the article.

A: 3.  We are afraid we do not exactly understand the reviewer’s suggestion. Does the reviewer suggest that we print some shape with the yeast cells? We do not understand how this will help understand the effects of the laser on yeast.

Round 2

Reviewer 1 Report

None.